# Microbiota Dysbiosis and Gut Barrier Dysfunction Associated with Non-Alcoholic Fatty Liver Disease Are Modulated by a Specific Metabolic Cofactors’ Combination

**DOI:** 10.3390/ijms232213675

**Published:** 2022-11-08

**Authors:** Sergio Quesada-Vázquez, Caitlin Bone, Shikha Saha, Iris Triguero, Marina Colom-Pellicer, Gerard Aragonès, Falk Hildebrand, Josep M. del Bas, Antoni Caimari, Naiara Beraza, Xavier Escoté

**Affiliations:** 1Eurecat, Technology Centre of Catalunya, Nutrition and Health Unit, 43204 Reus, Spain; 2Gut Microbes and Health Institute Strategic Programme, Quadram Institute Bioscience, Norwich Research Park, Norwich NR4 7UQ, Norfolk, UK; 3Food Innovation and Health Institute Strategic Programme, Quadram Institute Bioscience, Norwich Research Park, Norwich NR4 7UQ, Norfolk, UK; 4Nutrigenomics Research Group, Department of Biochemistry and Biotechnology, Universitat Rovira i Virgili, 43007 Tarragona, Spain; 5Digital Biology, Earlham Institute, Norwich Research Park, Norwich NR4 7UZ, Norfolk, UK; 6Eurecat, Centre Tecnològic de Catalunya, Biotechnology Area, 43204 Reus, Spain

**Keywords:** intestinal permeability, gut-liver axis, gut microbiota, metabolic disease, SCFAs

## Abstract

The gut is a selective barrier that not only allows the translocation of nutrients from food, but also microbe-derived metabolites to the systemic circulation that flows through the liver. Microbiota dysbiosis occurs when energy imbalances appear due to an unhealthy diet and a sedentary lifestyle. Dysbiosis has a critical impact on increasing intestinal permeability and epithelial barrier deterioration, contributing to bacterial and antigen translocation to the liver, triggering non-alcoholic fatty liver disease (NAFLD) progression. In this study, the potential therapeutic/beneficial effects of a combination of metabolic cofactors (a multi-ingredient; MI) (betaine, N-acetylcysteine, L-carnitine, and nicotinamide riboside) against NAFLD were evaluated. In addition, we investigated the effects of this metabolic cofactors’ combination as a modulator of other players of the gut-liver axis during the disease, including gut barrier dysfunction and microbiota dysbiosis. Diet-induced NAFLD mice were distributed into two groups, treated with the vehicle (NAFLD group) or with a combination of metabolic cofactors (NAFLD-MI group), and small intestines were harvested from all animals for histological, molecular, and omics analysis. The MI treatment ameliorated gut morphological changes, decreased gut barrier permeability, and reduced gene expression of some proinflammatory cytokines. Moreover, epithelial cell proliferation and the number of goblet cells were increased after MI supplementation. In addition, supplementation with the MI combination promoted changes in the intestinal microbiota composition and diversity, as well as modulating short-chain fatty acids (SCFAs) concentrations in feces. Taken together, this specific combination of metabolic cofactors can reverse gut barrier disruption and microbiota dysbiosis contributing to the amelioration of NAFLD progression by modulating key players of the gut-liver axis.

## 1. Introduction

The gut-liver axis is described as the compilation of events that take place among the liver, the gut, and gut microbiomes, which influence each other. This close connection underlines the critical regulatory effect of the gut microbiota on liver and gut health [1,2]. The liver is supplied with blood from the gut through the portal vein, which supports the close anatomical and functional interaction between gut microbiomes, the gut, and the liver [1]. This blood supply contains nutrients and (bacterial) metabolites needed for correct homeostasis [3,4]. When metabolic homeostasis is disrupted due to lifestyle deterioration in a westernized society, characterized by an unhealthy diet and reduced physical activity, this triggers multifactorial risk factors such as obesity, dyslipidemia, and metabolic syndrome [3], which could promote the hepatic manifestation known as non-alcoholic fatty liver disease (NAFLD). NAFLD is described as the most common liver disease, with a prevalence of around one-third of the world’s population and is still rising [1]. NAFLD is characterized by the accumulation of free fatty acids in the liver, called hepatic steatosis, which can progress to steatohepatitis and fibrosis in further stages.

The gut is a selective barrier that allows the translocation of nutrients and microbe-derived metabolites to the systemic circulation passing through the portal vein to the liver [3,4]. However, this barrier also effectively protects against the translocation of pathogenic bacteria and harmful microbe-derived products, such as bowel luminal antigens and inflammatory factors, through its selective permeability [1,4]. The intestinal epithelium has a self-renewing capacity during homeostasis and regenerates in response to injury via the proliferation of intestinal stem-cell-derived epithelial cells [5]. This intestinal barrier is controlled by tight-junctions (TJ) proteins and its expression and integrity are regulated by the immune system, which is molded by the microbiome composition [4], and by Occludin, Cadherin, and ZO-1 protein interactions [6].

Gut microbiota is composed of trillions of microorganisms that create a symbiotic relationship with the host or reside as commensals and can execute various functions influencing human physiology and pathology, such as the fermentation of indigestible fibers into short-chain fatty acids (SCFAs) that are crucial in some physiological processes [7,8,9,10]. When diets are imbalanced and contain excessive energetic and fatty intakes, intestinal microbiota diversity is disturbed, known as dysbiosis [1,3]. Dysbiosis has a critical impact on altering SCFA production, and on altered intestinal permeability, which is induced by TJ alteration that leads to epithelial barrier deterioration, and consequently bacterial products and antigen translocation to the liver associated with an elevated level of proinflammatory cytokines, which may lead to NAFLD development [11,12]. Intestinal microbiota dysbiosis has been connected to hepatic fat accumulation [13]. Thereby, the “leaky” gut hypothesis proposes the increment of energy harvest and altered choline metabolism by overnutrition increase intestinal permeability, which raise the production of proinflammatory mediators, and together with the alteration in SCFAs production due to gut microbiota dysbiosis, contributes to NAFLD pathogenesis [9,12,14].

Our recent studies have demonstrated [15,16] that the specific combination of metabolic cofactors composed of L-carnitine (LC; an enhancer of fatty acid uptake across the mitochondrial membrane), nicotinamide riboside (NR; NAD+ precursor), n-acetyl cysteine (NAC), and betaine (glutathione precursors and betaine a methyl donor) [17,18,19] is a promising treatment against NAFLD. This multi-ingredient (LC, NAC, Betaine, NR; hereinafter MI) supplementation improved pathological NAFLD features in the liver, reducing inflammation, steatosis, and insulin resistance [15]. Specifically, LC showed a boost effect on FFAs transport into the mitochondria in hepatocytes inducing lipid oxidation [20]; NAC was described to have a protective role in blocking hepatic steatosis and reducing proinflammatory mediators [21]; Khodayar et al. determined betaine is involved in methionine metabolism and may increase glutathione levels as an antioxidant action [22]; and NR can increase NAD^+^ levels and accelerate FFAs oxidation by SIRT1 activation and protecting against HFD-induced metabolic disorders [23]. However, the potential of this supplementation strategy in modulating the gut-microbiota-liver axis has not been explored. Thus, considering the strong connection between the liver, the gut, and the gut microbiota plays an important role in hepatic homeostasis and in the pathogenesis of different hepatic diseases [4,11], in the present study we demonstrate that the supplementation of a specific combination of metabolic cofactors in a preclinical model of NAFLD not only reverse NAFLD pathogenesis in the liver, but also ameliorates gut morphological changes, gut barrier permeability, and reduces intestinal inflammation by improving intestinal microbiota composition directly related to NAFLD development.

## 2. Results

### 2.1. Multi-Ingredient Treatment Modified Epithelium Morphology in the Small Intestine in a NAFLD

The lengths of the small intestine shortened significantly in NAFLD mice compared to control mice, but no significant effect was found after one month of MI treatment (Figure 1a). However, further histopathological analysis of different areas of the small intestine showed more evident differences when MI was supplemented by recovering intestinal morphology altered by NAFLD. As expected, control and NAFLD mice showed microscopic differences at the jejunum, with wider and shorter villus in NAFLD mice (Figure 1b). Villus length and width were increased in the NAFLD group compared to their counterparts (Figure 1c,d). In contrast, MI-supplemented mice significantly reduced both villus length and width compared to NAFLD mice (Figure 1c,d). Total intestinal wall thickness was increased in the NAFLD mice compared to the control group, while MI treatment significantly reduced the NAFLD effect on mice (Figure 1e). This effect was also translated both in the mucosa layer length (Figure 1f) and in the muscular layer length (Figure 1g), significantly ameliorating in the MI group compared to NAFLD mice. On the other hand, crypt depth was considerably reduced in NAFLD mice compared to the control group, and MI treatment could significantly increase the crypt depth (Figure 1h).

### 2.2. MI Supplementation Ameliorates Intestinal Permeability and Inflammation

Small intestine permeability was assessed in the jejunum by immunolocalization of Occludin, showing a significant reduction in apical staining in NAFLD mice compared to control mice, while Occludin apical expression was restored in the MI-supplemented group (Figure 2a,b). In accordance with the Occludin distribution, *Ocln* expression was downregulated in NAFLD mice compared to control mice. However, *Ocln* and *Cdh-1* mRNA expressions were up-regulated in the MI-supplemented group in comparison with the NAFLD mice group, and *Zo-1* tended to be up-regulated in the MI-supplemented group (Figure 2c).

In addition, NAFLD supplemented with the MI displayed a significant down-regulation in inflammation-related genes *Tnfα* and *Il-1β* expression levels when compared to their NAFLD counterparts, suggesting MI treatment influences the regulatory crosstalk between the immune system and the intestinal barrier function preserving intestinal permeability. In contrast, *Nlrp3* did not show changes in mRNA expression difference between groups (Figure 2d). In addition, intestinal *Cbs* expression levels (a key enzyme in GSH production to defend against oxidative stress) were downregulated in NAFLD animals (Figure 2d). However, animals treated with the MI supplementation reversed this downregulation by increasing intestinal *Cbs* expression levels similar to control animals (Figure 2d). Overall, our results support that MI treatment influences the regulatory crosstalk between the immune system and the intestinal barrier function preserving intestinal permeability.

### 2.3. MI Supplementation Recovers Proliferative Cells Localization in the Small Intestine

In NAFLD mice, the proliferative cell count and migration revealed a considerable reduction compared to the control groups (Figure 3a–c). Nevertheless, MI treatment increased the number of Ki-67-stained nuclei, both in the villus and crypts, which could mean that MI supplementation improves intestinal cell renewal in response to HF diets (Figure 3a–c).

Next, to assess intestinal mucosa function, it was determined *Klf4* mRNA expression, observing that it was increased in the MI supplementation group compared to NAFLD mice (Figure 3d). In addition, goblet cell count was significantly reduced in NAFLD mice compared to the control group (Figure 3e), while, in accordance with *Klf4* expression results, MI treatment significantly restored the number of goblet cells.

### 2.4. Microbiota Dysbiosis Present in NAFLD Is Attenuated after MI Supplementation

As expected, in the analysis of the bacterial diversity, NAFLD mice showed a tendency to decrease their diversity compared to control mice, but MI supplementation significantly increased bacterial diversity in comparison with NAFLD mice, suggesting that these metabolic cofactors increased gut microbiota diversity (Figure 4a). Further taxonomic analysis at the genus level showed substantial and significant differences between the study groups were observed (Figure 4b) [23]. NAFLD mice were characterized by a higher presence of *Anaerotruncus, Eubacterium nodatum, Lachnoclostridium, Lachnospiraceae UCG-001*, and *Escherichia/Shigella* in comparison to the control (Table 1). MI supplementation reduced the concentration of these bacteria (Table 1). Moreover, *Faecalibaculum, Christensenella, Faecalibacterium*, *Eggerthellaceae*, and *Enterococcus* were also increased in NAFLD mice in comparison with control mice, but MI treatment achieved a reduction in these genera similar to control levels. Interestingly, *Peptococcus*, *Butyricicoccus*, and *Ruminiclostridium* were only significantly increased in MI-treated mice compared to NAFLD and control mice (Table 1).

### 2.5. MI Supplementation Modulates the SCFAs Fecal Concentrations

In the analysis of SCFAs concentrations in feces, propionate was significantly increased in NAFLD mice compared to the control group, an effect that tended to be similar in the total fecal SCFAs (Figure 5). Interestingly, MI supplementation caused a significant reduction in propionate levels in feces from NAFLD mice. No important changes were found in other SCFAs detected (Figure 5).

### 2.6. Fecal SCFAs Levels Are Correlated with Changes in Specific Fecal Genera Bacteria

In the correlation analyses, changes in fecal propionate levels were positively correlated with the *Citrobacter* genus (Figure 6). However, after dividing the animals according to the treatments, only NAFLD mice reproduced this positive correlation of propionate levels with *Citrobacter*. *Eubacterium nodatum, Bacteroides, Anaerotruncus*, and *Faecalibacterium* showed a positive correlation with propionate, and *Bifidobacterium, Odoribacter, Lactobacillus, Lachnoclostridium, Lachnospiraceae UCG-001*, and *Faecalibaculum* were negatively correlated (Table 2). The control group found fecal propionate positively correlated with *Caproiciproducens* and *Ruminiclostridium* levels, and negatively with *Lactobacillus, Faecalibacterium*, and *Escherichia/Shigella* (Table 2). In contrast to NAFLD mice, MI supplementation promoted a negative correlation of fecal propionate with *Enterococcus*.

In addition, fecal acetate levels were negatively correlated with *Odoribacter, Enterococcus*, and *Lachnoclostridium* (Figure 6). In the control group, *Lactobacillus* and *Lachnoclostridium* were negatively correlated with acetate (Table 2). In NAFLD mice, *Bifidobacterium, Odoribacter Lactobacillus, Lachnoclostridum, Lachnospiraceae UCG-001,* and *Faecalibaculum* were negatively correlated with acetate. On the other hand, in NAFLD-MI mice, fecal acetate was not significantly correlated with any genera (Table 2).

Variations in fecal butyrate levels were not correlated with any genera (Figure 6). Nevertheless, when groups were arranged by treatment, some differences were found. In control mice, *Odoribacter* and *Lactobacillus* were negatively correlated with fecal butyrate levels, and *Caproiciproducens* was positively correlated. In NAFLD mice, *Eggerthellaceae, Ruminococcaceae UBA1819*, and *Christensenella* were positively correlated with butyrate, but *Romboutsia, Parasutterella*, and *Escherichia/Shigella* were negatively correlated.

Finally, iso-butyrate did not show any correlation with any bacterial genera (Figure 6). However, when groups were arranged by treatment, in the control group, only *Faecalibacterium* was negatively correlated with iso-butyrate. On the other hand, no correlations were found between iso-butyrate and any genera in NAFLD mice and MI-treated mice (Table 2).

## 3. Discussion

NAFLD is a multifactorial disease that involves different physio-pathological factors, including environmental, nutritional, genetics, epigenetics, and hormones. Furthermore, some of these factors also affect gut health, promoting intestinal barrier dysfunction and microbiota dysbiosis [1,9]. In turn, this gut impairment increases the development of NAFLD, disrupting the gut-liver axis [1,9]. Previous studies showed intestinal permeability, inflammation, dysbiosis, and gut morphology alterations in this hepatic disease [24,25,26,27]. Considering that NAFLD is a multifactorial pathology and that different studies have shown that better responses are obtained when treatments are performed under a multifaceted approach [17,28], the treatment of NAFLD using different bioactive compounds that act against complementary targets, such as the use of the aforementioned specific combination of metabolic cofactors, could be considered as a potential strategy to ameliorate NAFLD [15,16]. Here, we demonstrate that the MI treatment contributes to revert changes in the gastrointestinal tract linked to NAFLD development associated with ameliorating gut dysfunction and microbiota dysbiosis.

Gut morphology alterations are strongly related to high-caloric diets. Following prior studies, chronic ingestion of a high-fat diet elicits an increased length and width of villi in the jejunum, a decrease in crypt depth, and an increase in the total wall thickness and muscular and mucosa layers [29,30]. Here, it is observed that MI (LC, betaine, NAC, and NR) treatment relieved all these morphologic changes caused by diet-induced NAFLD, improving intestinal morphology to healthy levels. These features may be possible because NR increases intracellular NAD^+^ levels and may activate SIRT1, which has been related to an improvement in colony formation efficiency in crypts and intestinal stem cells [31]. Thus, villi elongation was reduced by NR supplementation linked to an inefficient cell renewal in an aging mice model that correlates with our results [31]. Another metabolic cofactor like betaine also had a protective effect on villus and crypts in a rat model, probably due to both the methyl group donor nature and the osmotic nature of betaine [32] and NAC, with protective effects because of its ability as a scavenger of oxygen free radicals; overall ameliorating histological injuries in the small intestine of rats [33].

A breach of the intestinal barrier caused by westernized diets allows the translocation of harmful bacteria and antigens, increasing liver damage and inflammation [3,34] which, if unresolved, can lead to the progression of NAFLD to advanced hepatic disease [3,35,36]. In this study, mice with NAFLD presented negative effects on key proteins in TJ integrity by reducing Occludin concentration and expression, and increased *Tnf-α* expression inducing inflammation, in accordance with a study with obese mice [37,38], while an improvement of the abnormal distribution of Occludin and *Zo-1* and *Cdh-1* expression and a reduced expression of inflammatory-related genes was induced by MI treatment in NAFLD mice. These results were correlated with a study that used LC supplementation, which reduced inflammatory biomarkers in the colon by modifying oxidative stress activity in a colitis animal model [39]. Interestingly, the increased expression of *Zo-1* and *Ocln* were recovered, and inflammatory biomarkers were decreased by betaine supplementation in acute liver failure mice [40]. In addition, the participation of NAC in modulating permeability and intestinal inflammation has also been described during in vitro and in vivo LPS-induced dysfunction by reducing oxidative stress and increasing *Zo-1*, *Cdh-1*, and *Ocln* gene expression [41]. Therefore, the capacity of the combination of these metabolic cofactors to reduce oxidative stress could diminish intestinal inflammation by ameliorating cytokine-mediated disruption of the intestinal TJ barrier, improving the expression and signaling of multiple integral proteins such as Occluding and Zo-1 [42]. Furthermore, to check the antioxidant effect of betaine and NAC as GSH precursors, *Cbs* expression, which is implicated in GSH and H_2_S biosynthesis [43], was analyzed. Intestinal *Cbs* expression was observed to be downregulated in NAFLD mice. The downregulation of *Cbs* expression could decrease H_2_S biosynthesis, which has anti-inflammatory and cytoprotective characteristics and could increase homocysteine levels, triggering inflammation in the small intestine [44,45], which connects with the upregulation of *Tnf-α* in this study. In contrast, MI supplementation recovered *Cbs* expression levels that could be related to an increased antioxidant response due to betaine and NAC supplementation, concordant with what has been observed in other tissues affected by NAFLD [15].

Regarding the importance of enterocyte-renewal mechanisms, cell proliferation has been described to be negatively influenced by HFDs [46]. Intracellular oxidative stress is induced by HFDs, predisposing the intestine to chronic diseases by disrupting cell proliferation [47]. In addition, proliferation was also affected by methyl donor deficiency, decreasing in the proximal small intestine mucosa [48]. In this study, disrupted cell proliferation was observed in NAFLD mice, closely related to the loss of intestinal permeability [49]. Further detrimental effects on intestinal cell function have been observed after HFDs, including reducing goblet cell differentiation and TJ alteration, which are critical for the correct mucus layer production and barrier permeability [41,46,50,51]. In addition, proliferative cells and goblet cell numbers in the intestinal epithelium were increased in MI-treated mice compared to NAFLD mice. This improvement may be due to the antioxidant properties of NAC and betaine. NAC offers protection against oxidative stress [41], and betaine is positively correlated with goblet cell differentiation [48] and can boost GSH levels and enhance the activity of antioxidant enzymes [52], as shown above.

The role of the microbiota during the pathogenesis of NAFLD has been widely studied. Differences in the gut bacterial composition may contribute to NAFLD development via different pathways, such as an increase in the expression levels of some pro-inflammatory cytokines in the liver [53]. As has been observed, decreased microbial diversity is associated with increased intestinal permeability and systemic low-grade inflammation [54], and consequently, linked to hepatic steatosis [9]. Importantly, the success in increasing intestinal bacterial diversity was caused by MI treatment, which may be linked to the beneficial effects found in the intestinal barrier and function. In addition, an increase in the *Firmicutes* levels at the phylum level was shown in NAFLD mice, followed by an increase in *Proteobacteria*, which was consistent with previous studies that demonstrated an increase in *Firmicutes* and *Proteobacteria* during NAFLD [26,55]. These changes in microbiota composition were reverted by MI supplementation, reducing the concentration of these phyla. These results were in accordance with previous studies that used betaine or NAC in HFD animal models to show reduced *Firmicutes* levels [40] and *Proteobacteria* levels [56], respectively. These observations were confirmed in a clinical study combining three of them (NR, NAC, and LC) [57], supporting the potential role of these compounds in modifying the gut bacterial community.

A deeper analysis of microbiota composition at the genus level showed that *Lachnospiraceae* increased in the NAFLD group, resembling the observations from a clinical study with NAFLD patients that found the levels of this genus significantly increased [58]. Moreover, *Anaerotruncus* and *Lachnoclostridium*, which are described as mucin-degrading bacteria that can impact both glycan composition and mucus thickness participating in the degradation of mucin [59], were increased in NAFLD mice in this study, which correlated with other preclinical studies [60,61], supporting the association of impaired intestinal barrier function with NAFLD. Here, we also found the elevated presence of *Eubacterium nodatum* in NAFLD mice, which is closely related to periodontal lesions and influences the pathology of NAFLD [62]. However, the levels of *Eubacterium nodatum* were significantly reduced by MI treatment. Furthermore, Yin et al. found an increased level of *Escherichia/Shigella* in rats with NAFLD, which are gram-negative bacteria containing LPS that may impair the gut barrier and trigger a low-grade chronic inflammation state [63], similarly to in NAFLD mice. Nevertheless, MI treatment succeeded in reducing *Escherichia/Shigella* levels. Furthermore, an elevation of the *Enterococcus* genus was presented in NAFLD mice in this study, a genus of potentially pathogenic bacteria with virulence factors and antibiotic resistance genes [64], which positively correlated with chronic liver diseases [65] and NAFLD [6]. Importantly, all these pathogenic bacteria related to mucin degradation, LPS-producers, or harmful bacteria were reduced after MI treatment. These changes in gut microbiota are associated with improved homeostasis and diversity of gut microbiota, connected with the actions of these metabolic cofactors in metabolic disorders shown separately in prior studies [40,56]. Finally, *Butyricicoccus*, which is a butyrate producer genus that reduces intestinal inflammation, was increased after MI supplementation. This fact is concordant with a previous study with NAFLD patients treated with metabolic cofactors [57]. In addition, the levels of *Peptococcus* were increased after MI supplementation, a similar situation was observed in HFD mice after naringin treatment [66], which correlates high *Peptococcus* abundance with an improvement in NAFLD progression.

A recent study emphasized that high fecal SCFAs content impact NAFLD progression by maintaining intestinal low-grade inflammation [67]. NAFLD and obese patients were characterized by high fecal propionate [67,68], corresponding with increased propionate levels [67,68], similar to the increase observed in NAFLD mice. Propionate is a key precursor for gluconeogenesis and lipogenesis, and inhibits lipolysis favoring lipid accumulation [68]. Importantly, propionate levels were reduced in NAFLD mice by MI supplementation, supporting the beneficial impact of this treatment in restoring gut-microbiota homeostasis. Furthermore, *Christensenella* levels, which were positively correlated with propionate levels [69] and increased in NAFLD [6], were elevated in NAFLD mice and reduced after MI supplementation. While previous studies showed that *Ruminiclostridium* negatively correlated with propionate [70], we here found elevated levels in MI mice, supporting the beneficial effects of this treatment. Considering that excessive propionate levels inevitably result in L-carnitine deficiency, the supplementation of L-carnitine may modulate propionate levels converting it into beneficial propionyl-carnitine and improving β-oxidation pathways [71]. Moreover, NAC also had a neuroprotective role against oxidative stress that is caused by propionate preserving GSH levels [71], which could be translated also into the gut after MI treatment. In addition, *Citrobacter* was positively correlated with propionate levels; Lee et al. found *Citrobacter* levels elevated according to fibrosis severity in NAFLD patients [72]. However, this positive correlation was exclusively found in NAFLD mice, but not in the control and MI groups. Therefore, the correlation of elevated levels of propionate with *Citrobacter* could be linked to NAFLD development.

There are some limitations of this observational study. First, the lack of use of single housing and paired-feeding techniques to control food intake individually in mice. However, social housing is essential for rodents, so housing them in individual cages is discouraged [73]. Second, conclusions derived from the present study were sustained in young male mice. Although this situation has commonly occurred in other studies [19,74,75], it is necessary to validate the effect of supplementation with MI in other models, both in older mice and in females. Third, the present study lacks an MI-treated group without NAFLD, but the study design was similar to other studies [19,75,76], and no deleterious effect was expected for this treatment.

In summary, this study successfully demonstrated that the specific combination of metabolic cofactors (NAC, NR, betaine, and LC) is a promising NAFLD nutraceutical, targeting gut-liver crosstalk disease and modulating gut dysfunction and dysbiosis. MI supplementation showed the capacity to recover beneficial bacteria levels and prevent harmful bacterial growth that, together with the restoration of the TJ barrier integrity, protects mice against proinflammatory bacterial product leakage and contributes to diminishing NAFLD development. In addition, MI supplementation induces the reduction of gut microbiota-derived propionate levels linked to decreased levels of Firmicutes contributing to the prevention of propionate-induced lipid accumulation in the liver. Therefore, this is the first study using MI supplementation as a potential treatment to treat gut dysfunction and microbiota dysbiosis associated with NAFLD in an animal model and could be a novel therapeutic strategy to ameliorate gut-liver crosstalk in NAFLD in clinical studies.

## 4. Materials and Methods

### 4.1. MI Treatment Composition

MI is a mix of the following compounds: 400 mg/kg of LC tartrate (Cambridge Commodities, Ely, UK), 400 mg/kg NAC (Cambridge Commodities), 800 mg/kg Betaine (Cambridge Commodities), and 400 mg/kg NR (ChromaDex, Los Angeles, CA, USA). LC was administrated through LC tartrate (LCT), containing 68.2% LC, and providing 560 mg/kg to reach the 400 mg LC/kg dose.

### 4.2. Animal Model and Experimental Design

Twenty-four C57BL/6J; 24 6-week-old male mice (Envigo, Barcelona, Spain) were housed in groups under controlled conditions of temperature (22 ± 2 °C) and humidity (55 ± 10%), and on a 12-h light/dark cycle with free access to food and water (Figure 7). After acclimatization, animals were randomly divided into experimental groups: Control mice (n = 8), kept on a standard diet (D12328, Research Diets, New Brunswick, NJ, USA), and a NAFLD group (n = 16), fed with a high-fat diet (HFHC: D12331, Research Diets) supplemented with 23.1 g/L fructose and 18.9 g/L sucrose in the drinking water. Mice were kept on these diets for a period of 20 weeks in ad libitum conditions [15]. For the last 4 weeks of the study (from the 16th to 20th week), NAFLD mice were randomly distributed into two groups: 8 mice were kept under the same fed conditions described above (NAFLD group), and 8 mice were exposed to multi-ingredient (MI) treatment (NAFLD-MI). Betaine, LCT, NAC, and NR were diluted with drinking water. Solutions were freshly prepared three times per week from stock powders and protected from light [15]. Fresh fecal pellets were rapidly collected 2 days before sacrifice and frozen at −80 °C. After 4 weeks of treatment, mice were sacrificed and small intestines were rapidly collected, measured, weighed, and divided into two sections (the first section was kept in formalin, and the other section was frozen in liquid nitrogen and stored at t −80 °C until further analysis) (Figure 7). All experimental protocols were approved by the Animal Ethics Committee of the Technological Unit of Nutrition and Health of Eurecat (Reus, Spain), and the Generalitat de Catalunya approved all the procedures (10281). The experimental protocol followed the “Principles of Laboratory Care” guidelines and was carried out in accordance with the European Communities Council Directive (2010/63/EEC).

### 4.3. Histological Staining Analysis of Intestinal Sections

To evaluate whether the MI treatment ameliorates intestinal dysfunction associated with NAFLD, physio-pathological features of the jejunum were analyzed. Small intestinal lengths were measured with a ruler. Jejunum sections were fixed in a 4% formaldehyde solution for 24 h and transferred to a 70% ethanol solution until paraffin inclusion. Tissue sections 4 µm thick were cut from paraffin blocks and placed on glass slides. Hematoxylin and eosin (H&E) staining was performed using standard procedures [77]. Small intestine images were taken with a microscope (ECLIPSE Ti; Nikon, Tokyo, Japan) coupled with a digital sight camera (DS-Ri1, Nikon) (10× magnification). To avoid any bias in the analysis, the study had a double-blind design, preventing the reviewers from knowing any data from the mice during the histopathological analysis. The morphometric analysis of the intestinal wall of the jejunum was conducted from these H&E sections. To evaluate villus height (distance from the villus-crypt junction to the top of the villus) and width (measured at the villus, distance between the villus-crypt junctions), crypt depth (depth of the invagination between adjacent villi), muscular layer, and mucosal layer (from the villus apex to the mesothelium of the tunica serosa), three measurements per every section of every animal were randomly made and analyzed using ImageJ NDPI software (National Institutes of Health, Bethesda, MD, USA; available at http://imagej.nih.gov/ij).

### 4.4. Immunofluorescence Analysis of Intestinal Sections

For intestinal permeability status, immunofluorescence using an anti-Occludin antibody (Abcam, Cambridge, UK) was used in paraffin cut sections of jejunum. A Cy3-labeled secondary antibody was used and slides were mounted in a DAPI-containing solution (Vector Laboratories, Burlingame, CA, USA). Three images per section were taken using a confocal immunofluorescent microscope, Zeiss LSM 800 Axio Observer (Zeiss, Thornwood, NY, USA) (magnification 40×), and fluorescence intensity was analyzed with Image J software. Fluorescence measured at the lumen of the colonic tissue was used as background fluorescence to subtract from the epithelial fluorescence values. Therefore, fluorescence density values represented the total protein in the epithelial cells [78].

### 4.5. Immunohistochemistry Analysis of Intestinal Sections

To evaluate epithelial cell proliferation in the small intestine after MI supplementation, Ki-67 immunohistochemistry (IHC) was performed on jejunum sections. Sample sections on slides were deparaffinized and hydrated through a descending scale of alcohol. Antigen retrieval was performed by boiling sections for 10 min in 5 mM sodium citrate buffer (pH 6, Sigma, Darmstadt, Germany) in a microwave oven. Endogenous peroxidase was blocked by incubation with 3% H_2_O_2_ (Sigma-Aldrich, St. Louis, MO, USA), 50% methanol solution (Sigma, Darmstadt, Germany) in PBS 13 (Lonza, Walkersville, MD, USA) for 20 min at room temperature (RT). The sections were then washed with PBS 1X and incubated for 30 min with rat anti-mouse Ki-67 (Dako, Glostrup, Denmark) diluted 1:50 in PBS, washed and incubated for 30 min RT with biotinylated secondary antibody rabbit anti-rat (Abcam, Cambridge, UK) diluted 1:100 in PBS. Sections were incubated with ABC-kit (Vector Laboratories, Burlingame, CA, USA) and 3,3-diaminobenzidine (DAB, Biocare Medical, Concord, CA, USA), counterstained with hematoxylin, dehydrated through an ascending scale of alcohols and xylene, and mounted with coverslips. All samples were observed and photographed (20× magnification) with a microscope, Olympus BX53, with a digital camera (Olympus Italia s.r.l., Segrate, Italy) and three fields for jejunum sections per mouse were analyzed to quantify proliferative cells by Image J software. Quantification of the number of goblet cells from one side of the villus was performed in these Ki-67-stained images.

### 4.6. Quantification of Short Chain Fatty Acids in Fecal Samples

Fresh fecal pellets were rapidly collected 2 days before sacrifice and frozen at −80 °C. SCFAs were extracted in 0.5% orthophosphoric acid, as outlined by Zhao et al. and García-Villalba et al. [79,80]. Samples were thawed on ice, centrifuged, and 10 µL of the supernatant was mixed with 90 µL 0.5% orthophosphoric acid containing all isotopically labeled internal standards (5 mM for acetate, 0.25 mM for lactate, and 0.5 mM for the other five). Samples were further centrifuged at 15,000 rpm for 10 min and the supernatant was transferred to the chromatography vials for analysis using LC-MS. For tissue samples, tissues were pulverized using a pestle and mortar under dry ice and mixed into a homogenous powder. An amount of 30 mg of each tissue was mixed with 200 µL of 0.5% orthophosphoric acid in water and homogenized using the Precellys 24 lysis homogenizer at 6000 rpm for two cycles for 30 s (Bertin Technologies, Montigny-le-Bretonneux, France). After centrifugation for 10 min at 4 °C, 45 µL of the supernatant was mixed with 5 µL of 0.5% orthophosphoric acid containing all isotopically labeled internal standards (0.5 mM for all except lactate which was 0.25 mM). Other sample extraction methods, including 100% methanol in 0.5% orthophosphoric acid (85% orthophosphoric acid diluted in methanol) and 50% methanol in 0.5% orthophosphoric acid, were also trialed, but these gave a cloudy mixture that did not completely separate even after centrifugation at 15,000 rpm for 15 min. Quantification of SCFAs by LC-MS/MS analysis was performed following the protocol previously described [81].

### 4.7. mRNA Extraction for Quantitative Polymerase Chain Reaction

Jejunums were homogenized for total mRNA extractions using TriPure reagent (Roche Diagnostic, Barcelona, Spain) according to the manufacturer’s instructions. mRNA concentration and purity were determined using a nanophotometer (Implen GmbH, München, Germany). mRNA was converted to cDNA using a High-Capacity RNA-to-cDNA Kit (Applied Biosystems, Wilmington, DE, USA). The cDNAs were diluted 1:10 before incubation with commercial LightCycler 480 Sybr green I master on a LightCycler^®^ 480 II (Roche Diagnostics GmbH, Manheim, Germany). Table 3 shows a list of used primers of those genes related to inflammation and intestinal mechanisms to be quantified with qPCR that were previously described in other studies and verified with Primer-Blast software (National Center for Biotechnology Information, Bethesda, MD, USA), using *36b4* as a housekeeping gene [82].

### 4.8. Bacterial Genomic DNA Isolation and 16s rRNA Sequencing

Bacterial genomic DNA was isolated from fecal pellets using an MBP DNA Soil extraction kit. Genomic DNA was normalized to 5 ng/µL with EB (10 mM Tris-HCl), and libraries were performed. Briefly, following a first PCR and clean-up using KAPA Pure Beads (Roche Catalogue No. 07983298001), a second PCR master mix was made up using P7 and P5 of Nextera XT Index Kit v2 index primers (Illumina Catalogue No. FC-131-2001 to 2004). Following the PCR reaction, the libraries were quantified using the Quant-iT dsDNA assay high-sensitivity kit (Catalogue No. 10164582) and run on a FLUOstar Optima plate reader. Libraries were pooled and run on a High Sensitivity D1000 ScreenTape (Agilent Catalogue No. 5067-5579) using the Agilent Tapestation 4200 to calculate the final library pool molarity. The pool was run on an Illumina MiSeq instrument using MiSeq^®^ Reagent Kit v3 (600 cycle) (Illumina Catalogue FC-102-3003) following the Illumina recommended denaturation and loading recommendations, which included a 20% PhiX spike in (PhiX Control v3 Illumina Catalogue FC-110-3001). The raw data was analyzed locally on the MiSeq using MiSeq reporter.

For the 16S sequence analysis, LotuS2 2.19 was used [88] in short read mode, using default quality filtering. Raw 16S rRNA gene amplicon reads were quality filtered to ensure: a minimum length of 170 bp; no more than eight homonucleotides; no ambiguous bases; average quality ≥ 27; and an accumulated read error < 0.5. Filtered reads were clustered using DADA2 [89] into Amplicon sequence variants (ASVs).

Postprocessing included de novo and reference-based chimera removal [90], back mapping dereplicated and mid-quality reads and off-target removal [91], resulting in 1005 ASVs and 4.2Mil out of 5.7Mil read pairs being used in the final analysis. The taxonomy to ASVs was assigned by calculating each ASVs least common ancestor using the LCA of LotuS2 and a mapping against SILVA 138.1 [92]. Abundance matrices were normalized using RTK [93].

### 4.9. Statistical Analysis

Statistical analyses were performed using GraphPad Prism 9 software (Graph-Pad Software, La Jolla, CA, USA). Data were presented as mean ± SEM. Data distribution was analyzed by the Shapiro–Wilk normality test. Differences between the two groups were determined using an unpaired t-test (two-tailed, 95% confidence interval). One-way analysis of variance (ANOVA) was conducted to examine the differences between the three groups.

Alpha diversity was analyzed using Shannon’s diversity index for the bacterial community regarding operational taxonomic units (OTUs) [24].

In addition, 16s rRNA sequencing data analysis was conducted with R statistical language Version 3.00 (The R Foundation; available at https://www.r-project.org/) as described in Hildebrand et al. [94], employing the RTK software [93] or all data normalizations. Statistical differences between multiple samples at Phylum and Genera levels were estimated by Kruskal-Wallis or Mann–Whitney U-test, or Negative Binomial distribution [95] by adjusting for multiple testing according to the method of Benjamini and Hochberg (q-value) [96]. A summary of differences in genus abundance between the different animal groups is listed in Table 1, when the *p*-value < 0.05.

To investigate further into potential interactions between the SCFAs levels and microbiota diversity, correlation analyses were performed. Origin 8.1 Pro software (OriginLab, Northampton, MA, USA) was used for the Spearman correlation analysis. The genera plotted were those significant in the 16s rRNA sequencing analysis between groups, and those with significant correlations with SCFAs. A *p*-value below 0.05 was considered statistically significant.

## Figures and Tables

**Figure 1 ijms-23-13675-f001:**
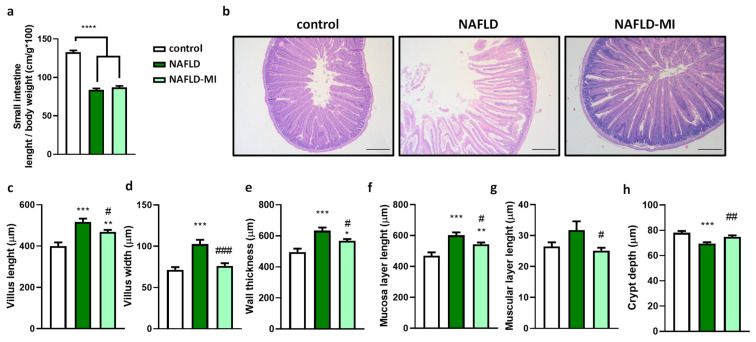
(**a**) Relative length of the small intestine of control, NAFLD, and NAFLD-MI groups. (**b**) Hematoxylin and eosin (H&E) staining of jejunums. Bar = 200 µm. Morphometric intestinal variables associated with NAFLD: (**c**) villus length, (**d**) villus width, (**e**) wall thickness, (**f**) mucosa layer length, (**g**) muscular layer length, and (**h**) crypt depth. Data are mean ± SEM. * *p* < 0.05 vs. control mice; ** *p* < 0.01 vs. control mice; *** *p* < 0.001; **** *p* < 0.0001 vs. control mice. # *p* < 0.05 vs. NAFLD mice; ## *p* < 0.01 vs. NAFLD mice; ### *p* < 0.001 vs. NAFLD mice.

**Figure 2 ijms-23-13675-f002:**
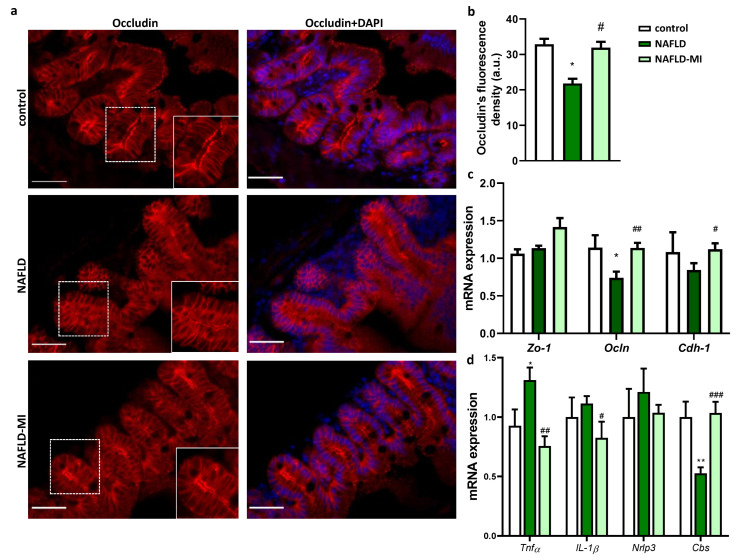
**Effects of the MI treatment on intestinal permeability and inflammation.** (**a**) Jejunum immunofluorescence of Occludin and DAPI. Bar = 50 µm. (**b**) Occludin’s fluorescence intensity. Effects of treatments on (**c**) mRNA expression of intestinal permeability-related genes: *Zo-1* (Zonula Oclnudens-1), *Ocln* (Occludin), and *Cdh-1* (Cadherin-1), on (**d**) mRNA expression of inflammatory and antioxidant related-genes: *Tnfα* (Tumor Necrosis Factor α), *Il-1β* (interleukin 1β), *Nlrp3* (NLR family pyrin domain containing 3), and *Cbs* (cystathionine-β-synthase). Data are mean ± SEM. * *p* < 0.05, ** *p* < 0.01 vs. control mice, # *p* < 0.05; ## *p* < 0.01; ### *p* < 0.001 vs. NAFLD mice.

**Figure 3 ijms-23-13675-f003:**
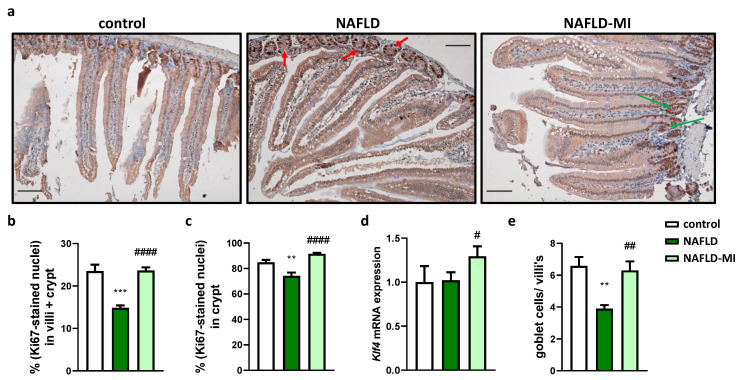
**NAFLD mice showed an ectopic localization of proliferative cells which is recovered after MI supplementation.** (**a**) Immunohistochemical analysis of small intestine sections of proliferating cells with Ki-67. Red arrows show the reduction of proliferative cells in crypts in NAFLD compared to the control group. Green arrows show the recovery of proliferation cells’ localization after MI supplementation. Moreover, in MI-supplemented mice and control mice, proliferative cells were detected in the villus. Bar = 200 µm. (**b**) Percentage of ki-67-stained nuclei in villus and crypts compared to other types of cells, and (**c**) the percentage of ki-67-stained nuclei just in crypts. Improvement of mucosa function was evaluated by intestinal mRNA expression of *Klf4* (Kruppel-like factor 4) (**d**,**e**) goblet cells count. Data are mean ± SEM. ** *p* < 0.01, *** *p* < 0.001 vs. control mice; # *p* < 0.05, ## *p* < 0.01, #### *p* < 0.0001 vs. NAFLD mice.

**Figure 4 ijms-23-13675-f004:**
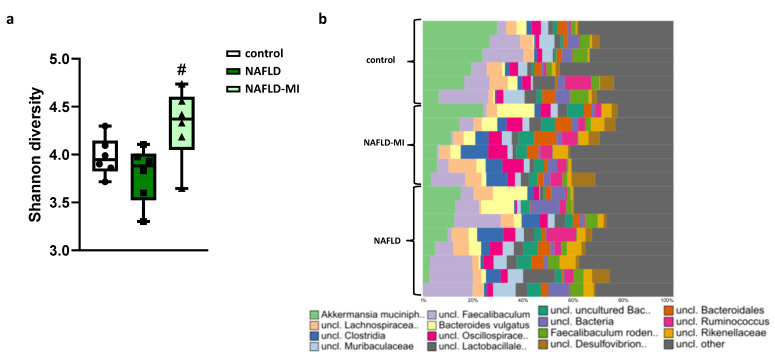
(**a**) Estimation of bacterial diversity as assessed by the Shannon index. (**b**) Overview of microbial composition at genus level in the control, NAFLD, and NAFLD-MI groups. # *p* < 0.05 vs. NAFLD mice.

**Figure 5 ijms-23-13675-f005:**
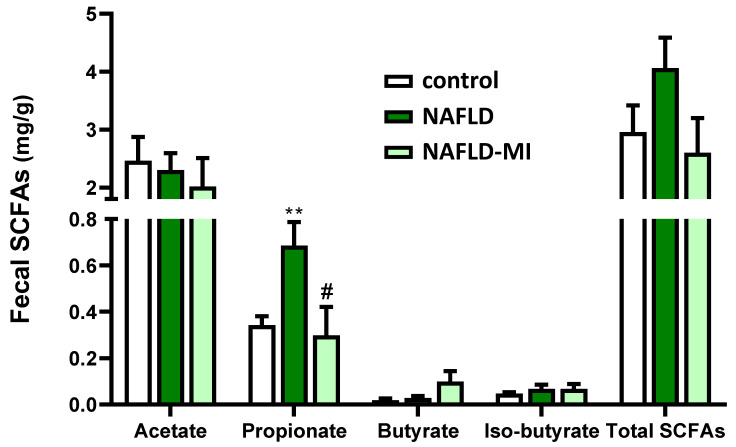
SCFAs quantification in fecal samples. Fecal amounts of the different short-chain fatty acids were analyzed in control, NAFLD, and NAFLD-MI mice. Data are mean ± SEM. ** *p* < 0.01 vs. control mice; # *p* < 0.05 vs. NAFLD mice.

**Figure 6 ijms-23-13675-f006:**
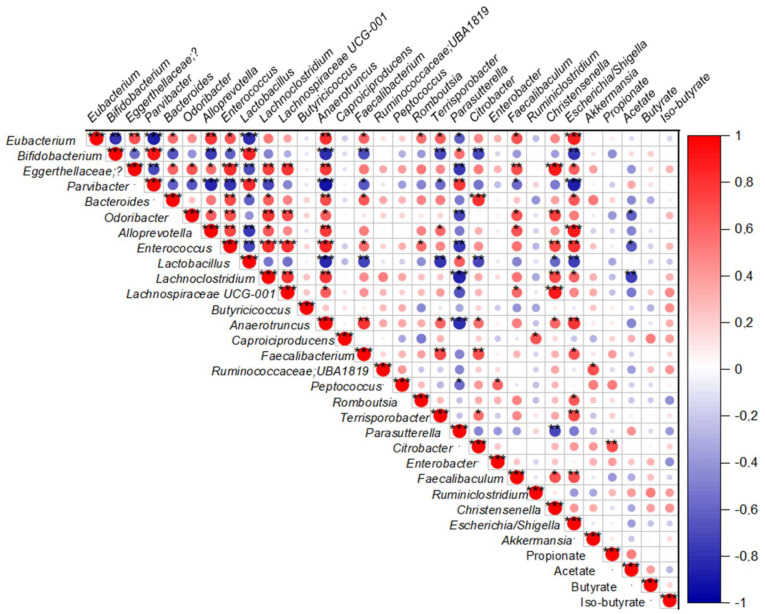
Spearman correlation analysis of fecal SCFAs and selected fecal genus from all mice. Dots in red mean positive correlation, whereas dots in blue mean negative correlation. * *p* < 0.05, ** *p* < 0.01, *** *p* < 0.001.

**Figure 7 ijms-23-13675-f007:**
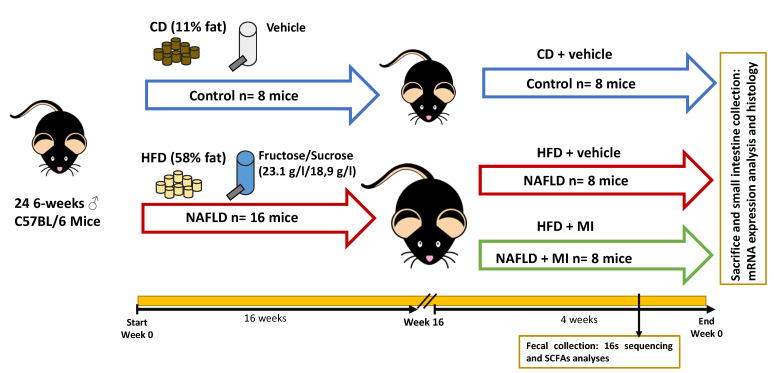
Schematic representation of the study design for the induction of NAFLD in mice (n = 24 animals; n = 8 control mice and n = 16 NAFLD mice), and the following treatment with MI supplementation in NAFLD mice for 4 weeks (n = 8 control mice, n = 8 NAFLD mice, and n = 8 NAFLD-MI mice). The variables analyzed in all study groups are represented. Abbreviations: CD, control-chow diet; HFD, high-fat diet; NAFLD, non-alcoholic fatty liver disease; SCFAS, short chain fatty acids.

**Table 1 ijms-23-13675-t001:** Differences in genus abundance between the different animal groups.

Testing against Control, NAFLD, and MI; Requested Test(s): Wilcox; Used Matrix Type: Normed; Used Test Statistic Is: Kruskal-Wallis Test. Only Genera Are Listed When the *p*-Value < 0.05.Genus	Phylum	Direction	*p*-Value	q-Value
** *Anaerotruncus* **	*Firmicutes*	NAFLD > MI >> control	0.0008	0.0291
** *Butyricicoccus* **	*Firmicutes*	*MI > control = NAFLD*	*0.0036*	*0.0515*
** *Christensenella* **	*Firmicutes*	*NAFLD > MI = control*	*0.0350*	*0.0920*
** *Eggerthellaceae* **	*Actinobacteria*	*NAFLD > MI = control*	*0.0139*	*0.0679*
** *Enterococcus* **	*Firmicutes*	*NAFLD > control = MI*	*0.0065*	*0.0557*
** *Eubacterium nodatum* **	*Firmicutes*	NAFLD > MI > control	0.0011	0.0317
** *Escherichia/Shigella* **	*Proteobacteria*	NAFLD > MI > control	0.0015	0.0374
** *Faecalibaculum* **	*Firmicutes*	NAFLD > MI = control	0.0270	0.0920
** *Faecalibacterium* **	*Firmicutes*	NAFLD > MI = control	0.0167	0.0753
** *Lachnoclostridium* **	*Firmicutes*	NAFLD > MI > control	0.0040	0.05
** *Lachnospiraceae UCG-001* **	*Firmicutes*	NAFLD > MI > control	0.0065	0.05
** *Peptococcus* **	*Firmicutes*	MI > control = NAFLD	0.0349	0.0920
** *Ruminiclostridium* **	*Firmicutes*	MI > control = NAFLD	0.021	0.088

**Table 2 ijms-23-13675-t002:** Spearman correlation analysis of selected fecal genus bacteria and SCFAs in all animals, control, NAFLD, and NAFLD-MI groups. Boxes in red mean positive correlation and boxes in blue mean negative correlation. * *p* < 0.05, ** *p* < 0.01, *** *p* < 0.001.

		*Eubacterium*	*Bifidobacterium*	*Eggerthellaceae;?*	*Parvibacter*	*Bacteroides*	*Odoribacter*	*Alloprevotella*	*Enterococcus*	*Lactobacillus*	*Lachnoclostridium*	*Lachnospiraceae UCG-001*	*Butyricicoccus*	*Anaerotruncus*	*Caproiciproducens*	*Faecalibacterium*	*Ruminococcaceae;UBA1819*	*Peptococcus*	*Romboutsia*	*Terrisporobacter*	*Parasutterella*	*Citrobacter*	*Enterobacter*	*Faecalibaculum*	*Ruminiclostridium*	*Christensenella*	*Escherichia-Shigella*	*Akkermansia*
**Propionate**	All animals																					**						
	control									*					*									*	*		*	
	NAFLD	***	***			***	***			***	***	***		***		***						***		***				
	NAFLD-MI								***																			
**Acetate**	All animals						*		*		**																	
	control									*	*																	
	NAFLD	***	***			***	***			***	***	***		***		***						***		***				
	NAFLD-MI																											
**Butyrate**	All animals																											
	control						*			**					**													
	NAFLD			***													***		***		***					***	***	
	NAFLD-MI																											
**Iso-butyrate**	All animals																											
	control																							*				
	NAFLD																											
	NAFLD-MI																											
						Positive correlation					Negative correlation																	

**Table 3 ijms-23-13675-t003:** Sequences of the oligonucleotides used in the RT-PCR.

Primers	Forward	Reverse	Reference
*Cdh-1*	5′-CATCCCAGAACCTCGAAACA-3′	5′-TGGGTTAGCTCAGCAGTAA-3′	This study
*Cbs*	5′-GCAGCGCTGTGTGGTCATC-3′	5′-CATCCATTTGTCACTCAGGAACTT-3′	[15]
*Il-1* *β*	5′-GGACCCCAAAAGATGAAGGGCTGC-3′	5′-GCTCTTGTTGATGTGCTGCTGCG-3′	[83]
*Klf4*	5′-AGCCACCCACACTTGTGACTATG-3′	5′-CAGTGGTAAGGTTTCTCGCCTGTG-3′	[50]
*Nlrp3*	5′-GCCCAAGGAGGAAGAAGAAG-3′	5′-AGAAGAGACCACGGCAGAAG-3′	[84]
*Ocln*	5′-ACCCGAAGAAAGATGGATCG-3′	5′-CATAGTCAGATGGGGGTGGA-3′	[85]
*Ost* *α*	5′-CACTGGCTCAGTTGCCATTT-3′	5′-GCATACGGCATAAAACGAGGT-3′	[86]
*Tnf* *α*	5′-AGGGTCTGGGCCATAGAACT-3′	5′-CCACCACGCTCTTCTGTCTAC-3′	[87]
*Zo-1*	5′-TGGGAACAGCACACAGTGAC-3′	5′-GCTGGCCCTCCTTTTAACAC-3′	[85]
*36b4*	5′-AGTCCCTGCCCTTTGTACACA-3′	5′-CGATCCGAGGGCCTCACTA-3′	[82]

## Data Availability

The data that support the findings of this study are available from the corresponding author upon reasonable request.

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
