# Peer review of "Microbiota Dysbiosis and Gut Barrier Dysfunction Associated with Non-Alcoholic Fatty Liver Disease Are Modulated by a Specific Metabolic Cofactors’ Combination"

_ijms, 2022, doi:10.3390/ijms232213675_

Round 1

Reviewer 1 Report

REVIEW

Microbiota dysbiosis and gut barrier dysfunction associated with NAFLD are modulated by a specific metabolic cofactors’ combination.

The present study is focused on the efficacy of a multi-ingredient mixture in NAFDL. This is a preclinical study in mice. The study is well designed with many interesting variables analyzed and demonstrates the efficacy of the intervention product. However, there are points for improvement in explaining the results. I also advise a serious revision of the language and the writing.

TITLE

·         Abbreviations in the title such as NAFLD should be avoided.

ABSTRACT

·         All abbreviations appearing in the abstract must be specified.

INTRODUCTION

·         Throughout the text in general and in the introduction in particular it seems that the terms microbiota and microbiome are used as synonyms, and they are not. The term microbiome is broader. Check throughout the text how these terms are used.

·         Line 61: Not all bacteria in the gut are in symbiosis with our organism, many of them are commensals. Please check the sentence.

MATERIAL AND METHODS

·         For a better understanding of the paper, it would be necessary that the material and methods section follows the introduction.

·         4.1 Animal Model and Diets:

o   “The animal study design was detailed in [13]”: Develop it in this paper and reference it, but this sentence is not acceptable.

o   The characteristics of the intervention product should be included in a separate section, as this section describes the animal model.

o   Describe what was done with the mice at the end of the intervention period (sacrifice). State that this animal model was reviewed and approved by an ethics committee.

o   A figure illustrating the study design as well as the variables analysed in all study groups would be appropriate.

o   “Fresh fecal pellets were rapidly collected 2 days before sacrifice and frozen at -80 â—¦C”: This sentence does not seem to apply very appropriate here. There should be a section on the design of the study and a general description of all the studies carried out on the mice.

·         Line 423: Please describe H&E.

·         4.2 Histological analysis, immunohistochemistry, and immunofluorescence of intestinal sections: It would be more appropriate to divide this section into three sections with more specific names.

·         Why is RNA extracted from the mouse intestine and then converted to DNA and not directly extracted DNA? Specify which genes are intended to be quantified with this qPCR.

·         4.6 Statistical Analysis:

o   Line 534: When you are beginning to develop the statistics of the data obtained from the 16S sequencing, this should go in another paragraph.

o   Line 541: Metagenomic analysis? NGS of the bacterial 16S rRNA gene is performed but no metagenomic studies are performed, I don't know what you mean by this. Please rephrase the sentence.

o   The p-value does not indicate the strength of association, so to speak of a p-value of less than 0.1 as a trend is not appropriate. Correct the figures in this respect.

RESULTS

·         In the results section there are several parts that should not be included here but in the material and methods section (this section should be more detailed to better understand the study). In the results section only the results of the study should be included, without assessments or additions:

o   To evaluate whether the MI treatment ameliorates intestinal dysfunction associated with NAFLD, physio-pathological features of the jejunum were analyzed.

o   Occludin and ZO-1 interactions modulate intestinal TJs integrity [22].

o   In addition, mRNA expression of some genes related to inflammation was performed in the jejunum sections.

o   The intestinal epithelium has self-renewing capacity during homeostasis and regenerates in response to injury via the proliferation of intestinal stem-cell-derived epithelial cells [23].

o   Alpha diversity was analyzed using Shannon’s diversity index for the bacterial community (Figure 4a) regarding operational taxonomic units (OTUs) [24].

o   SCFAs are direct metabolites made from the fermentation of dietary fiber and resistant starch by gut microbiota [25]. Changes in gut microbiota composition influence SCFAs production.

o   To deeper into potential interactions between the SCFAs levels and the microbiota diversity, correlation analyses were performed.

·         Figure 1: c to h figures. Resolution needs to be improved.

·         Line 171: Faecalibaterium is repeated.

·         Figure 3: b to e figures. It does not appear which bar each group belongs to.

·         Table 1: Incorrectly placed in the text. No mention of q-values in material and methods.

·         Lines 130-138: There is no mention in the material and methods that the transcriptome of genes related to inflammation in jejunal sections will be studied.

·         Figure 5: It does not appear which bar each group belongs to.

·         Lines 235-246: Develop each SFCA in separate paragraphs.

DISCUSSION

·         It is recommended that the discussion be written in passive rather than active form.

·         What are the limitations of the study?

·         Are these results translatable to humans? Are there clinical trials with positive results with similar products?

·         The conclusions are not a compilation of the most important results. This is already clear from the first paragraph of the discussion. In the last paragraph of the discussion, you should state what your conclusions are, based on the results discussed above.

SUPPLEMENTARY MATERIAL

·         Can be incorporated into the main manuscript.

Author Response

Reviewer 1 (round 1)

Microbiota dysbiosis and gut barrier dysfunction associated with NAFLD are modulated by a specific metabolic cofactors’ combination.

The present study is focused on the efficacy of a multi-ingredient mixture in NAFDL. This is a preclinical study in mice. The study is well designed with many interesting variables analyzed and demonstrates the efficacy of the intervention product. However, there are points for improvement in explaining the results. I also advise a serious revision of the language and the writing.

Thank you so much for your kindly constructive and professional suggestions. We really appreciate all your comments that have helped to improve our manuscript. We really apologize for the errors detected in the manuscript.

Below you will find the answer to your comments highlighted in red.

TITLE

  • Abbreviations in the title such as NAFLD should be avoided.

RESPONSE (R): As it has been indicated by the reviewer, this point has been amended.

ABSTRACT

  • All abbreviations appearing in the abstract must be specified.

R: As it has been indicated by the reviewer, this point has been amended.

INTRODUCTION

  • Throughout the text in general and in the introduction in particular it seems that the terms microbiota and microbiome are used as synonyms, and they are not. The term microbiome is broader. Check throughout the text how these terms are used.

R: We agree with the reviewer's assessment, and microbiota and microbiome were placed in their correct term throughout the text.

  • Line 61: Not all bacteria in the gut are in symbiosis with our organism, many of them are commensals. Please check the sentence.

R: We are grateful for the suggestion and the sentence was changed in the new version of the manuscript. Line 65:

“Gut microbiota is composed of trillions of microorganisms that create a symbiotic relationship with the host and or reside as commensals and can execute various functions influencing human physiology and pathology, such as the fermentation of indigestible fibers into short-chain fatty acids (SCFAs) that are crucial in some physiological processes”

MATERIAL AND METHODS

  • For a better understanding of the paper, it would be necessary that the material and methods section follows the introduction.

R: We agree with your suggestion, which provably allow a better understanding of the article: However, we have followed the Guidelines of the IJMS for the authors while preparing the manuscript.

  • 4.1 Animal Model and Diets:
  • “The animal study design was detailed in [13]”: Develop it in this paper and reference it, but this sentence is not acceptable.

R: We agree with your suggestion including the following paragraph. Line 975:

For the last 4 weeks of the study (from the 16th to 20th week), NAFLD mice were randomly distributed into two groups: 8 mice were kept under the same fed conditions described above (NAFLD group), and 8 mice were exposed to multi-ingredient (MI) treatment (NAFLD-MI). Betaine, LCT, NAC, and NR were diluted with drinking water. Solutions were freshly prepared three times per week from stock powders and protected from light [15]. Fresh fecal pellets were rapidly collected 2 days before sacrifice and frozen at -80 â—¦C. After 4 weeks of treatment, mice were sacrificed and small intestines were rapidly collected, measured, weighed, and divided into two sections (the first section was kept in formalin and the other section was frozen in liquid nitrogen and stored at −80 â—¦C until further analysis (Figure 7).”

  • The characteristics of the intervention product should be included in a separate section, as this section describes the animal model.

R: We agree with your suggestion, and it was included in a new different section. Line 961:

4.1 MI treatment composition

MI is a mix of the following compounds: 400 mg/kg of LC tartrate (Cambridge Commodities, Ely, UK), 400 mg/kg NAC (Cambridge Commodities), 800 mg/kg Betaine (Cambridge Commodities) and 400 mg/kg NR (ChromaDex, Los Angeles, CA, USA). LC was administrated through LC tartrate (LCT), containing 68.2 % LC, providing 560 mg/kg to reach the 400 mg LC/kg dose. “

  • Describe what was done with the mice at the end of the intervention period (sacrifice). State that this animal model was reviewed and approved by an ethics committee.

R: Following the reviewer suggestion, this information was added. Line 981:

After 4 weeks of treatment, mice were sacrificed and small intestines were rapidly collected, measured, weighed, and divided into two sections (the first section was kept in formalin and the other section was frozen in liquid nitrogen and stored at t −80 â—¦C until further analysis (Figure 7). All experimental protocols were approved by the Animal Ethics Committee of the Technological Unit of Nutrition and Health of Eurecat (Reus, Spain) and the Generalitat de Catalunya approved all the procedures (10281). The experimental protocol followed the “Principles of Laboratory Care” guidelines and was carried out in accordance with the European Communities Council Directive (2010/63/EEC)”

  • A figure illustrating the study design as well as the variables analysed in all study groups would be appropriate.

R: We agree with the reviewer suggestion and a new figure illustrating the study design and the variables analyzed was included (Figure 7).

Line 990:

Figure 7. Schematic representation of the study design for the induction of NAFLD in mice (n = 24 animals; n = 8 control mice and n = 16 NAFLD mice), and the following treatment with MI supplementation in NAFLD mice for 4 weeks (n = 8 control mice, n = 8 NAFLD mice and n = 8 NAFLD-MI mice). The variables analysed in all study groups are represented. Abbreviations: CD, control-chow diet; HFD, high fat diet; NAFLD, non-alcoholic fatty liver disease; SCFAS, short chain fatty acids.

  • “Fresh fecal pellets were rapidly collected 2 days before sacrifice and frozen at -80 â—¦C”: This sentence does not seem to apply very appropriate here. There should be a section on the design of the study and a general description of all the studies carried out on the mice.

R: We agree with your suggestion, and it was added in the animal model and experimental design section. Line 436

  • Line 423: Please describe H&E

R: The meaning of H&E was explained in the text, and it was briefly explained an H&E staining standard protocol. Line 457:

“Tissue sections 4 µm thick were cut from paraffin blocks and placed on glass slides. Hematoxylin and eosin (H&E) staining was performed using standard procedures [78].”

  • 4.2 Histological analysis, immunohistochemistry, and immunofluorescence of intestinal sections: It would be more appropriate to divide this section into three sections with more specific names.

R: We agree with your suggestion, and sections were separated in these following sections.

Line 453: 4.3 Histological staining analysis of intestinal sections

Line 471: 4.4 Immunofluorescence analysis of intestinal sections

Line 481: 4.5 Immunohistochemistry analysis of intestinal sections

  • Why is RNA extracted from the mouse intestine and then converted to DNA and not directly extracted DNA? Specify which genes are intended to be quantified with this qPCR.

R: We appreciate your question. For RT-qPCR analysis, mRNAs were extracted and then treated with DNase to remove the impurities from the genomic DNA that could be amplified in the qPCR, where we want to measure the specific expression of genes. After that, reverse transcription was performed for the mRNA samples to obtain cDNAs that will be amplified in the qPCR. In addition, the pairs of genes used have been paired in contiguous exons to avoid the amplification of contaminating genomic DNA.

    4.6 Statistical Analysis:

  • Line 534: When you are beginning to develop the statistics of the data obtained from the 16S sequencing, this should go in another paragraph.

R: We agree with your suggestion, and the sentence was split. Line 557:

For the 16S sequence analysis, LotuS2 2.19 was used [90] in short read mode, using default quality filtering. Raw 16S rRNA gene amplicon reads were quality filtered to ensure a minimum length of 170 bp, not more than eight homonucleotides, no ambiguous bases, average quality >= 27 and an accumulated read error < 0.5. Filtered reads were clustered using DADA2 [91] into Amplicon sequence variants (ASVs).”

  • Line 541: Metagenomic analysis? NGS of the bacterial 16S rRNA gene is performed but no metagenomic studies are performed, I don't know what you mean by this. Please rephrase the sentence.

R: Following the reviewer recommendation, the sentence was rephrased as follows. Line 585:

“To deeper into potential interactions between the SCFAs levels and microbiota diversity, correlation analyses were performed. Origin 8.1 Pro software (OriginLab, USA) was used for the Spearman correlation analysis. The genera plotted were those significant in the 16s rRNA sequencing analysis between groups, and those with significant correlations with SCFAs. A p-value below 0.05 was considered statistically significant.”

  • The p-value does not indicate the strength of association, so to speak of a p-value of less than 0.1 as a trend is not appropriate. Correct the figures in this respect.

R: We agree with your suggestion, and the indicated figures were amended.

RESULTS

  • In the results section there are several parts that should not be included here but in the material and methods section (this section should be more detailed to better understand the study). In the results section only the results of the study should be included, without assessments or additions:
  • To evaluate whether the MI treatment ameliorates intestinal dysfunction associated with NAFLD, physio-pathological features of the jejunum were analyzed.

R: We agree with your suggestion and this sentence was added in section “4.3 Histological staining analysis of intestinal sections”, line 454.

  • Occludin and ZO-1 interactions modulate intestinal TJs integrity [22].

R: We agree with the reviewer suggestion and the sentence was rephrased and added in Introduction section. Line 63:

This intestinal barrier is controlled by tight-junctions (TJ) proteins and its expression and integrity are regulated by the immune system, which is molded by the microbiome composition [4], and by Occludin, Cadherin and ZO-1 proteins interactions [6].”

  • In addition, mRNA expression of some genes related to inflammation was performed in the jejunum sections.

R: The sentence was rephrased and added in Material and Methods section, in the subsection “4.7 mRNA Extraction for Quantitative Polymerase Chain Reaction section”. Line 528:

Table 3 shows a list of used primers of those genes related to inflammation and intestinal mechanisms to be quantified with qPCR that were previously described in other studies and verified with Primer-Blast software (National Center for Biotechnology Information, Bethesda, MD, USA), using 36b4 as a housekeeping gene [83].”

  • The intestinal epithelium has self-renewing capacity during homeostasis and regenerates in response to injury via the proliferation of intestinal stem-cell-derived epithelial cells [23].

R: We agree with your suggestion and this sentence was incorporated in the Introduction section, Line 59.

  • Alpha diversity was analyzed using Shannon’s diversity index for the bacterial community (Figure 4a) regarding operational taxonomic units (OTUs) [24].

R: We agree with your suggestion and the sentence was included in the Material and Methods section, in the subsection “4.9 Statistical Analysis”. Line 576.

  • SCFAs are direct metabolites made from the fermentation of dietary fiber and resistant starch by gut microbiota [25]. Changes in gut microbiota composition influence SCFAs production.

R: As suggested by the reviewer, this sentence was eliminated in the new manuscript version.

  • To deeper into potential interactions between the SCFAs levels and the microbiota diversity, correlation analyses were performed.

R: We agree with your suggestion and the sentence was added in the Material and methods section in 4.9 Statistical Analysis section. Line 586

  • Figure 1: c to h figures. Resolution needs to be improved.

R: Graphs were enlarged with bigger font size increasing the resolution.

  • Line 171: Faecalibateriumis repeated.

R: Actually, this word is not repeated, they are different genera, for one side Faecalibaculum and the other one Faecalibacterium.

Figure 3: b to e figures. It does not appear which bar each group belongs to.

R: We agree with your suggestion, and a legend indicating the three different groups was incorporated in the figure 3.

  • Table 1: Incorrectly placed in the text. No mention of q-values in material and methods.

R: We agree with your suggestion, and q-values were mentioned in the Material and Methods section, in the “4.9 Statistical Analysis” subsection, as follows. Line 580:

      “Statistical differences between multiple samples at Phylum and Genera-level were estimated by Kruskal-Wallis or Mann–Whitney U-test or Negative Binomial distribution [98] by adjusting for multiple testing according to the method of Benjamini and Hochberg (q-value) [99].”

  • Lines 130-138: There is no mention in the material and methods that the transcriptome of genes related to inflammation in jejunal sections will be studied.

R: We agree with your suggestion and the genes studied in mRNA expression are listed in Table 3. Moreover, to clarify that inflammatory-related genes were analyzed in jejunal sections, we added the following sentence. Line 527:

Table 3 shows a list of used primers of those genes related to inflammation and intestinal mechanisms to be quantified with qPCR that were previously described in other studies and verified with Primer-Blast software (National Center for Biotechnology Information, Bethesda, MD, USA), using 36b4 as a housekeeping gene [83].”

  • Figure 5: It does not appear which bar each group belongs to.

R: As pointed above, a legend indicating the three different groups was incorporated in the figure 5.

  • Lines 235-246: Develop each SFCA in separate paragraphs.

R: Following the reviewer suggestion, SCFAs are explained in different paragraphs in the Results section.

 DISCUSSION

  • It is recommended that the discussion be written in passive rather than active form.

R: We agree with your suggestion and some actives sentences were changed to passive form in the Discussion section.

  • What are the limitations of the study?

R: We agree with your suggestion and the limitations of the study were included. Line 395:

“There are some limitations of this observational study. First, the lack of use of single housing and paired-feeding techniques to control food intake individually in mice. However, social housing is essential for rodents, so housing them in individual cages is discouraged [74]. Second, conclusions derived from the present study are sustained in young male mice. Although this situation occurs commonly in other studies [19,75,76], it is necessary to validate the effect of supplementation with MI in other models, both in older mice and in females. Third, the present study lacks an MI-treated group without NAFLD, but the study design was similar to other studies [19,76,77] and no deleterious effect was expected for this treatment.”

  • Are these results translatable to humans? Are there clinical trials with positive results with similar products?

R: We appreciate your questions. Some clinical trials that used some ingredients from the MI combination with positive results are referenced inside the text. These specific doses were determined based on previous studies and a calculation of dose translation from human to animal dosage as is explained in the study of Reagan-Shaw et. al., to obtain a comparable study to humans. All ingredients were diluted in drinking water and given ad libitum to the mice. Hence, it is specified in the conclusions that it could be a novel therapeutic strategy to ameliorate gut-liver crosstalk in NAFLD in clinical studies because can be translatable to humans.

Reagan-Shaw, S.; Nihal, M.; Ahmad, N. Dose Translation from Animal to Human Studies Revisited. FASEB J. 2008, 22, 659–661

  • The conclusions are not a compilation of the most important results. This is already clear from the first paragraph of the discussion. In the last paragraph of the discussion, you should state what your conclusions are, based on the results discussed above.

R: We agree with your suggestion and the conclusions were changed as follows. Line 404:

In summary, in this study is successfully demonstrated that the specific combination of metabolic cofactors (NAC, NR, betaine, and LC) is a promising NAFLD nutraceutical targeting gut-liver crosstalk disease and modulating gut dysfunction and dysbiosis. MI supplementation showed the capacity to recover beneficial bacteria levels and prevent harmful bacterial growth that, together with the restoration of the TJs barrier integrity, protects mice against proinflammatory bacterial product leakage and contributes to diminish NAFLD development. In addition, MI supplementation induces the reduction of gut microbiota-derived propionate levels linked to decreased levels of Firmicutes contributing to the prevention of propionate-induced lipid accumulation in the liver. Therefore, this is the first study using MI supplementation as a potential treatment to treat gut dysfunction and microbiota dysbiosis associated with NAFLD in an animal model and could be a novel therapeutic strategy to ameliorate gut-liver crosstalk in NAFLD in clinical studies

 SUPPLEMENTARY MATERIAL

  • Can be incorporated into the main manuscript.

R: We agree with your suggestion and Table 2 was incorporated into the novel version of the main manuscript.

Reviewer 2 Report

Comments for the Author (Required):

The study by Quesada-Vázquez S et al, titled “Microbiota dysbiosis and gut barrier dysfunction associated with NAFLD are modulated by a specific metabolic cofactors’ combination aims to study the effect of metabolic cofactors on NAFLD. Although this is an interesting article, I have some concerns about this article. 

Figure 2: Author showed that Occludin expression is lower in NAFLD when compared to control and MI supplementation group. It would be nice to show ZO -1 protein and claudin-1 protein in the respective group. 

It would be nice to quantify RNA for Occludin and Claudin-1. 

Author claims that the inflammation in NAFLD. However, there is no significant difference in TNF-alpha and IL-1b between control and NAFLD group. Please clarify.  

It would be interesting to check antioxidant enzyme activity in all the group. 

Minor 

Typo error- author mentioned Nrlp3 in many places- it should be Nlrp3

Author Response

Reviewer 2

The study by Quesada-Vázquez S et al, titled “Microbiota dysbiosis and gut barrier dysfunction associated with NAFLD are modulated by a specific metabolic cofactors’ combination” aims to study the effect of metabolic cofactors on NAFLD. Although this is an interesting article, I have some concerns about this article. 

Thank you so much for your kindly constructive and professional suggestions. We really appreciate all your comments that have helped to improve our manuscript. We really apologize for the errors detected in the manuscript.

Below you will find the answer to your comments highlighted in red.

Figure 2: Author showed that Occludin expression is lower in NAFLD when compared to control and MI supplementation group. It would be nice to show ZO -1 protein and claudin-1 protein in the respective group. 

It would be nice to quantify RNA for Occludin and Claudin-1. 

RESPONSE (R): We agree with your suggestion. Unfortunately, it is not possible performing this protein analysis because the samples were exhausted for these additional protein analyses.

However, other studies used the mRNA expression levels of these in gut to validate their functions (Wu et al. 2020). Following the reviewer suggestions, it was performed an mRNA expression analysis of Ocln and Cdh-1 genes to analyze the mRNA expression of the tight-junctions-related genes and evaluate the improvement of intestinal permeability. These results have been included in the new version of the article (Figure 2d) and in the Results section, as follows (Line 122):

In accordance with the Occludin distribution, Ocln expression was downregulated in NAFLD mice compared to control mice. However, Ocln and Cdh-1 mRNA expression were up-regulated in the MI-supplemented group in comparison with the NAFLD mice group, and Zo-1 tended to increase in the MI-supplemented group. (Figure 2c).”

Wu, J., He, C., Bu, J. et al. Betaine attenuates LPS-induced downregulation of Occludin and Claudin-1 and restores intestinal barrier function. BMC Vet Res 16, 75 (2020). https://doi.org/10.1186/s12917-020-02298-3

Author claims that the inflammation in NAFLD. However, there is no significant difference in TNF-alpha and IL-1b between control and NAFLD group. Please clarify. 

R: We agree with your suggestion and TNFα expression graph was reanalyzed with more samples and corrected, and the NAFLD group showed a significant up-regulation of TNFα compared to the control group, which agrees with what is written in the results and discussion.

It would be interesting to check antioxidant enzyme activity in all the group. 

R: We agree with your suggestion but, as commented above, samples are exhausted to perform this kind of antioxidant enzyme activity analysis. However, we performed an mRNA expression analysis of Cbs, a key enzyme in the GSH biosynthesis pathway, which promotes protection against oxidative stress (Werge et al. 2021).

It was observed an important downregulation of the Cbs expression in the NAFLD group. In contrast, after the MI-treatment, Cbs expression levels were recovered similarly to the control animals, indicating potential protection against oxidative stress.

These results are included in Figure 2d and in the results section as follows (Line 138):

“In addition, intestinal Cbs expression levels (a key enzyme in GSH production to defend against oxidative stress) were downregulated in NAFLD animals (Figure 2d). However, animals treated with the MI supplementation reversed this downregulation by increasing intestinal Cbs expression levels similar to control animals (Figure 2d).”

Werge, M.P. et al. The Role of the Transsulfuration Pathway in Non-Alcoholic Fatty Liver Disease. J. Clin. Med. 2021, 10, 1081. https://doi.org/10.3390/jcm10051081 Minor 

Typo error- author mentioned Nrlp3 in many places- it should be Nlrp3.

R: This mistake has been amended throughout the manuscript.

Round 2

Reviewer 1 Report

OK

Reviewer 2 Report

The manuscript has been revised and improved accordingly to the reviewers comments.